# Systemic Lupus Erythematosus: Ophthalmological Safety Considerations of Emerging and Conventional Therapeutic Agents

**DOI:** 10.3390/ijms262311744

**Published:** 2025-12-04

**Authors:** Wojciech Luboń, Małgorzata Luboń, Anna Agaś-Lange, Mariola Dorecka

**Affiliations:** 1Department of Ophthalmology, Faculty of Medical Sciences, Medical University of Silesia, 40-514 Katowice, Poland; mdorecka@sum.edu.pl; 2Department of Ophthalmology, Professor K. Gibiński University Clinical Center, Medical University of Silesia, 40-514 Katowice, Poland; kozikowskamalg@gmail.com (M.L.);

**Keywords:** systemic lupus erythematosus, ocular adverse events, targeted therapies, biologic agents, immunomodulation, ophthalmological safety

## Abstract

Systemic lupus erythematosus (SLE) is a chronic multisystem autoimmune disorder in which ocular involvement represents a clinically significant yet frequently underrecognized contributor to morbidity. Ocular manifestations in SLE may arise from disease activity itself, but also as adverse effects of long-term pharmacological therapy. With the advent of targeted immunomodulatory agents, the therapeutic landscape of SLE has expanded beyond conventional drugs such as hydroxychloroquine and corticosteroids toward biologics and small molecules designed to interfere with specific immunological pathways. These advances have improved systemic disease control and survival; however, their ophthalmological safety profiles remain only partially defined. This review synthesizes current evidence on ocular adverse events associated with both traditional and emerging SLE therapies. Established agents, particularly hydroxychloroquine and corticosteroids, are consistently linked to complications including retinopathy, posterior subcapsular cataracts, steroid-induced glaucoma, and central serous chorioretinopathy. In contrast, recently approved or investigational therapies—such as belimumab, anifrolumab, voclosporin, dual BAFF/APRIL inhibitors, rituximab, JAK inhibitors, CD40/CD40L blockade, CD38 inhibition, and mesenchymal stromal cell-based strategies—have limited but evolving safety data, with potential ocular adverse events spanning inflammatory, vascular, neuro-ophthalmic, and structural domains. Although ocular complications appear infrequent in clinical trials, underdetection in real-world practice and insufficient long-term monitoring may underestimate their true incidence. These findings highlight the need for systematic ophthalmological surveillance in patients receiving immunomodulatory therapies for SLE. Early recognition and timely management of ocular toxicity are crucial to safeguarding visual function and optimizing long-term therapeutic outcomes in this vulnerable patient population.

## 1. Introduction

Systemic lupus erythematosus (SLE) is a chronic, heterogeneous autoimmune disease characterized by multisystem involvement and considerable morbidity. Its pathogenesis arises from a complex interplay of genetic susceptibility, epigenetic modifications, and environmental triggers that converge on a dysregulated immune response [1]. Aberrant activation of both innate and adaptive immunity—particularly type I interferon signaling, autoreactive B cells, and T-cell dysfunction—drives chronic inflammation and immune complex deposition, ultimately leading to multi-organ damage [1].

Clinical manifestations of SLE are protean, affecting the skin, musculoskeletal system, kidneys, cardiovascular system, central nervous system, and frequently the eyes. Ocular involvement is reported in approximately one-third of SLE patients, manifesting in a broad clinical spectrum. The most common ocular manifestation is keratoconjunctivitis sicca, while the more vision-threatening complications include retinal vasculitis, optic neuritis, and ischemic optic neuropathy [2,3]. If undetected or inadequately managed, these ocular complications may result in irreversible visual impairment, substantially reducing quality of life and contributing to long-term disability. Importantly, ocular manifestations may serve not only as local complications but also as markers of systemic disease activity, reflecting underlying vascular and immunological dysregulation [4].

Traditionally, treatment of SLE has relied on corticosteroids and conventional immunosuppressants. Hydroxychloroquine (HCQ), a cornerstone of maintenance therapy, significantly improves long-term outcomes by reducing disease flares, thrombotic risk, and damage accrual. However, it carries a well-documented risk of retinal toxicity, particularly with prolonged exposure, higher cumulative doses, and patient-specific risk factors such as renal impairment [2]. Similarly, systemic corticosteroids remain indispensable in controlling severe flares, yet their ocular side effects—including posterior subcapsular cataracts, steroid-induced glaucoma, and central serous chorioretinopathy—pose important limitations [5].

Over the past decade, the therapeutic landscape of SLE has shifted dramatically with the introduction of targeted biologics and small molecules. Belimumab, a monoclonal antibody targeting B-cell activating factor (BAFF), was the first biologic approved for SLE, demonstrating efficacy in reducing disease activity and preventing flares [1]. More recently, anifrolumab, an antagonist of the type I interferon receptor, has shown benefit in patients with moderate-to-severe and refractory disease [1]. Voclosporin, a next-generation calcineurin inhibitor, has been approved for lupus nephritis and offers improved pharmacokinetics compared with cyclosporine. In addition, a diverse pipeline of investigational therapies—including telitacicept (a dual BAFF/APRIL inhibitor), anti-CD20 antibodies such as rituximab, CD40/CD40L modulators, CD38 inhibitors, Janus kinase (JAK) inhibitors, and mesenchymal stromal cell (MSC)-based therapies—reflects the ongoing diversification of therapeutic strategies [5].

While these advances expand treatment opportunities and improve systemic disease control, their ophthalmological safety profiles remain incompletely characterized. Clinical trials and real-world data have begun to reveal a spectrum of ocular complications. Reported manifestations include microvascular attenuation detectable by optical coherence tomography angiography (OCTA), inflammatory conditions such as episcleritis and uveitis, and opportunistic infections related to immunosuppression [6,7]. The unique immune-privileged environment of the eye may render it particularly susceptible to unintended effects of immune modulation, raising concerns about ischemic, inflammatory, and neuro-ophthalmic sequelae. Furthermore, subtle microvascular and neurodegenerative changes detectable by OCTA may precede overt clinical symptoms, suggesting that ocular imaging could serve as a non-invasive biomarker for subclinical drug toxicity as well as systemic disease activity [6].

Despite the increasing availability of safety data, several knowledge gaps persist. Most pivotal clinical trials in SLE do not systematically capture ophthalmological endpoints, and post-marketing surveillance often underestimates rare but clinically significant adverse events. Real-world data are fragmented, and the long-term ophthalmic safety of novel immunotherapies remains insufficiently studied. This paucity of comprehensive safety evaluations underscores the need for integrated ophthalmological monitoring in patients receiving both conventional and targeted therapies.

The purpose of this review is therefore threefold: (1) to synthesize available evidence on ocular adverse events associated with conventional, biologic, and small-molecule therapies in SLE; (2) to highlight emerging ophthalmological safety signals across different drug classes; and (3) to identify critical gaps in current knowledge that warrant further investigation. By addressing these objectives, this work emphasizes the importance of considering ocular outcomes as key safety endpoints in both clinical practice and future therapeutic trials. In doing so, it also provides an updated and integrative perspective by systematically comparing conventional and emerging therapies within a unified ophthalmological safety framework. Unlike prior reviews that focused primarily on clinical manifestations or single drug classes, this work combines mechanistic, clinical, and safety-related insights to offer a broader understanding of how immunomodulatory treatments may influence ocular health. 

## 2. Results

This review identified a broad spectrum of therapeutic strategies for systemic lupus erythematosus (SLE), encompassing both established immunotherapies and emerging biologic agents. Evidence was drawn from randomized controlled trials, long-term extension studies, and real-world observational data, with particular emphasis on ocular safety profiles. Findings are organized by therapeutic class, beginning with B-cell targeted agents and interferon pathway inhibitors, and extending to calcineurin inhibitors, dual BAFF/APRIL blockade, conventional immunotherapies, and novel investigational approaches such as JAK inhibitors, anti-CD40/anti-CD40L therapies, anti-CD38 antibodies, and mesenchymal stromal cell-based treatments.

Comparative discussion with hydroxychloroquine and corticosteroids highlights the spectrum of well-established ocular risks, providing a reference point against which newer therapies can be contextualized. Subsequent subsections provide a detailed synthesis of evidence across major therapeutic classes, emphasizing both systemic efficacy and ophthalmologic safety considerations.

### 2.1. B-Cell Targeted Therapy: Belimumab

Belimumab, a fully human monoclonal antibody directed against the B-cell activating factor (BAFF), was the first biologic therapy approved for systemic lupus erythematosus. By inhibiting BAFF, belimumab reduces abnormal B-cell survival and autoantibody production, thereby decreasing disease activity and flare frequency. Phase III randomized controlled trials and long-term extension studies consistently confirmed its clinical efficacy, including in patients with lupus nephritis, and demonstrated a favorable long-term safety profile [8,9,10,11,12,13].

Across pooled analyses of controlled clinical trials, the incidence of adverse events (AEs) in belimumab-treated patients was comparable to placebo (92.8% vs. 92.0%), with serious adverse events (SAEs) occurring in 15.2% vs. 17.0%, respectively [8].

The most common AEs included upper respiratory tract infections (22–25%), headache (14–17%), and gastrointestinal complaints such as nausea (11–15%) [8,9]. Serious infections were reported in approximately 5% of patients, most often pneumonia [10]. The discontinuation rate due to AEs remained low, ranging from 5 to 7% across studies [11,12].

With regard to ocular safety, large-scale clinical trial datasets and meta-analyses have not identified belimumab as a consistent cause of ophthalmological adverse events. No increased incidence of uveitis, episcleritis, or retinopathy has been documented in controlled settings [8,14]. Nevertheless, ocular involvement is common in SLE itself, including keratoconjunctivitis sicca, retinopathy, optic neuropathy, and anterior uveitis, often related to immune-complex deposition and vascular injury. While isolated case reports have described ocular complications in patients receiving belimumab, causality remains unproven. Theoretically, BAFF blockade could influence ocular immune homeostasis through altered B-cell regulation and downstream cytokine signaling.

In clinical practice, ophthalmological complications are not expected in patients treated with belimumab. However, increased vigilance may be advisable in individuals with pre-existing ocular comorbidities. Until larger pharmacovigilance datasets become available, careful ophthalmologic monitoring in high-risk groups remains prudent. In summary, belimumab demonstrates robust efficacy and an overall favorable safety profile in SLE, with no strong evidence of direct ocular toxicity, but systematic surveillance is warranted.

### 2.2. Type I Interferon Pathway Inhibition: Anifrolumab

Anifrolumab is a fully human monoclonal antibody targeting the type I interferon receptor subunit 1 (IFNAR1), thereby blocking the signaling of all type I interferons. Since type I interferon overexpression is a central mediator of SLE pathogenesis and correlates with disease activity, this pathway represents an attractive therapeutic target. In pivotal phase III clinical trials (TULIP-1 and TULIP-2), anifrolumab significantly reduced global disease activity and facilitated glucocorticoid tapering in patients with moderate-to-severe SLE [15,16].

Integrated analyses of phase II and III trials demonstrated a generally favorable safety profile. Although the overall incidence of adverse events was high (>90%), rates were comparable between treatment and placebo groups. Serious adverse events occurred in 11–16% of patients [15]. The most common events were upper respiratory tract infections (22–27%), nasopharyngitis (15–18%), bronchitis (10–12%), and infusion-related reactions (8–10%). Herpes zoster occurred more frequently in anifrolumab-treated patients (~5–7% vs. 2–3% with placebo), consistent with the established role of interferon signaling in antiviral defense [17,18]. Long-term extension studies confirmed durable efficacy and revealed no new safety signals, with discontinuation rates due to adverse events ranging from 6% to 8% [18,19,20,21].

Importantly, no consistent ocular adverse events—such as chorioretinitis, neuroretinitis, or uveitis—have been reported in clinical trials or integrated safety analyses. While type I interferons contribute to retinal immune privilege and antiviral protection, current clinical evidence does not support a causal relationship between anifrolumab and ocular inflammation. Nevertheless, the theoretical risk of opportunistic viral reactivation at immune-privileged sites, including the retina, justifies enhanced pharmacovigilance. Incorporating dedicated ophthalmological endpoints into future studies would strengthen understanding of the long-term safety profile.

In summary, anifrolumab is an effective and generally safe therapeutic option for moderate-to-severe SLE. Its adverse event profile is dominated by respiratory infections and herpes zoster, while ocular complications remain unsubstantiated by clinical trial evidence. However, continued vigilance in real-world and post-marketing settings is recommended, particularly in patients with pre-existing ocular comorbidities.

### 2.3. Calcineurin Inhibition: Voclosporin

Voclosporin is a next-generation calcineurin inhibitor approved for the treatment of lupus nephritis. It shares mechanistic similarities with cyclosporine and tacrolimus but offers improved pharmacokinetic stability and reduced interpatient variability. In the pivotal phase III AURORA 1 trial, voclosporin in combination with mycophenolate mofetil and low-dose glucocorticoids significantly increased the rate of complete renal response compared with placebo (41% vs. 23%, *p* < 0.0001) [22]. Long-term extension data from the AURORA 2 trial confirmed sustained efficacy and a manageable safety profile with up to three years of exposure [23].

The most frequently reported adverse events included gastrointestinal symptoms (diarrhea in 19–23%), hypertension (14–20%), headache (12–16%), and decreased renal function, consistent with calcineurin pathway inhibition [22,24]. Serious adverse events occurred in approximately 21% of voclosporin-treated patients, similar to placebo, with discontinuations due to adverse events in 7–9% [22,25]. Meta-analyses and post-marketing pharmacovigilance studies have supported these findings, with infections and renal function decline representing the most common causes of discontinuation [26,27].

Unlike traditional calcineurin inhibitors, large randomized controlled trials with voclosporin have not consistently reported ocular adverse events such as elevated intraocular pressure, optic disc edema, or posterior subcapsular cataracts. These complications, however, are biologically plausible given the established ocular toxicities of cyclosporine and tacrolimus [22,28]. Experimental evidence links calcineurin inhibition to altered trabecular meshwork function, impaired aqueous humor outflow, and posterior lens changes, all of which could predispose to glaucoma or cataract formation.

Moreover, systemic consequences of calcineurin blockade—particularly hypertension and renal impairment—may indirectly increase ocular vulnerability by exacerbating microvascular stress and accelerating retinal or optic nerve damage. Post-marketing surveillance has not yet identified a clear safety signal, but underreporting of ophthalmological endpoints remains a limitation [27].

Taken together, while direct ocular toxicity has not been confirmed with voclosporin, individualized risk stratification remains essential. Patients with pre-existing glaucoma, ocular hypertension, or lens pathology may benefit from periodic ophthalmological monitoring, particularly during long-term therapy. Incorporating structured ophthalmic assessments (OCT, fundus imaging, IOP monitoring) in future trials would help to clarify the true ocular safety profile of this agent.

### 2.4. Dual BAFF/APRIL Inhibition: Telitacicept and Atacicept

Telitacicept and atacicept are investigational fusion proteins that simultaneously inhibit B-cell activating factor (BAFF) and a proliferation-inducing ligand (APRIL). Both cytokines are critical for B-cell survival, plasma cell differentiation, and immunoglobulin production. Dual blockade of BAFF and APRIL is hypothesized to more effectively suppress autoreactive B-cell activity compared with selective BAFF inhibition alone.

Clinical studies of telitacicept have demonstrated encouraging efficacy in SLE, with phase II and real-world trials showing significant reductions in disease activity and glucocorticoid use. In a multicenter phase IIb trial, response rates (SRI-4) at week 48 were significantly higher with telitacicept compared to placebo (71% vs. 37%, *p* < 0.001) [29]. The overall incidence of adverse events exceeded 80%, but most were mild to moderate, including upper respiratory tract infections (15–25%), nasopharyngitis (10–18%), headache (8–12%), and injection-site reactions (7–10%) [30,31,32]. Serious adverse events occurred in 8–12% of patients, with rates comparable to placebo, and discontinuation rates were low (≤7%). Long-term observational studies have confirmed a favorable safety profile without cumulative toxicity [13,33].

Atacicept has been studied in phase II clinical trials for SLE and lupus nephritis. While biologically effective, its development was limited by safety concerns, including hypogammaglobulinemia and increased susceptibility to serious infections, which led to early trial discontinuations [34].

Importantly, neither telitacicept nor atacicept has been associated with consistent ocular adverse events in controlled studies. Reports of blurred vision or nonspecific visual disturbances remain anecdotal and lack systematic ophthalmological assessment. However, given the central role of BAFF/APRIL pathways in systemic immune regulation, subclinical ocular effects cannot be excluded. In particular, impaired B-cell regulation could theoretically alter ocular immune privilege, disrupt retinal microvascular stability, or increase susceptibility to opportunistic ocular infections.

The absence of structured ophthalmological endpoints in existing trials limits interpretation of these potential risks. Therefore, future studies should incorporate objective ophthalmic measures—such as spectral-domain optical coherence tomography (SD-OCT), optical coherence tomography angiography, and retinal microvasculature imaging—to ensure comprehensive evaluation of long-term ocular safety.

### 2.5. Conventional Immunotherapies: Hydroxychloroquine and Corticosteroids

Hydroxychloroquine (HCQ) remains a cornerstone in systemic lupus erythematosus maintenance therapy due to its disease-modifying properties, flare-preventive effects, and long-term survival benefits. However, retinal toxicity is a well-documented, dose- and duration-dependent adverse event. The classical manifestation, bull’s-eye maculopathy, results from parafoveal photoreceptor and retinal pigment epithelium damage. The risk of retinopathy rises significantly when daily doses exceed 5 mg/kg real body weight or after cumulative exposure beyond five years [35,36,37]. Large epidemiological analyses estimate a prevalence of definite toxicity of 7.5% after ≥5 years of treatment, increasing to >20% after 20 years [38]. Retinal damage is typically irreversible and may progress despite cessation of therapy, highlighting the importance of preventive screening.

Advances in diagnostic imaging, including SD-OCT, multifocal electroretinography, and automated visual field testing, have markedly improved sensitivity for early detection [39,40]. The American Academy of Ophthalmology (AAO) revised its guidelines in 2011 and 2016, recommending baseline screening within the first year of HCQ initiation and annual examinations after five years, or earlier in high-risk patients [37]. Nevertheless, adherence to these guidelines in clinical practice remains inconsistent, with population studies showing compliance rates as low as 30–40% [41,42].

Systemic corticosteroids, a mainstay in SLE for acute flares and organ-threatening manifestations, are associated with multiple ocular complications. Posterior subcapsular cataracts (PSC) are the most frequent, with risk correlating with both cumulative dose and duration; prevalence rates range from 20% to 40% among long-term steroid users [43]. Steroid-induced glaucoma is another major concern, driven by increased trabecular meshwork resistance and impaired aqueous outflow. Approximately 18–36% of the general population are considered “steroid responders” with clinically significant intraocular pressure (IOP) elevation during therapy [44,45]. In addition, central serous chorioretinopathy (CSCR) is strongly associated with exogenous corticosteroid exposure, with an odds ratio >4 compared with non-exposed controls [46].

Collectively, these ocular complications may result in irreversible vision loss if undetected. Therefore, routine ophthalmological evaluations are a critical component of long-term management in SLE patients treated with HCQ or corticosteroids. Preventive strategies include strict HCQ dose monitoring, adherence to AAO screening recommendations, and regular intraocular pressure and lens assessments in patients receiving systemic corticosteroids.

### 2.6. Anti-CD20 Therapy: Rituximab

Rituximab, a chimeric monoclonal antibody directed against CD20 on B cells, has been used off-label in refractory systemic lupus erythematosus for more than two decades. Although phase II/III randomized controlled trials (EXPLORER and LUNAR) failed to meet their primary efficacy endpoints, numerous observational studies and registry analyses support its utility in severe, treatment-resistant cases, including lupus nephritis and neuropsychiatric lupus [47,48,49,50].

The safety profile of rituximab is dominated by infusion-related reactions (20–30% of patients), hypogammaglobulinemia, and increased susceptibility to infections. Serious infections occur in approximately 10–15% of patients, and opportunistic viral reactivations, such as hepatitis B and JC virus, have been reported [47].

Ocular adverse events are not consistently described in randomized trials. However, case reports and small case series document viral retinitis, particularly cytomegalovirus (CMV) and herpes simplex retinitis, in patients with profound immunosuppression [51]. These complications are biologically plausible, as B cells and antibody-mediated immunity are essential for the control of latent viral infections within ocular tissues.

Thus, rituximab is not intrinsically associated with direct ocular toxicity, but secondary infection-related retinitis should be considered in heavily immunosuppressed patients. Routine ophthalmological monitoring is not required for all patients; however, targeted evaluation is advisable in those presenting with visual symptoms or carrying additional risk factors for viral reactivation.

### 2.7. Anti-CD40 Therapy: BI 655064

I 655064 is a fully human antagonistic monoclonal antibody targeting CD40, a costimulatory molecule expressed on antigen-presenting cells that interacts with CD40 ligand (CD40L/CD154) on T cells. Disruption of this pathway downregulates T-cell activation and B-cell help, thereby reducing autoantibody production and proinflammatory cytokine release. Given the pivotal role of aberrant T–B cell crosstalk in systemic lupus erythematosus (SLE), CD40 blockade represents a promising therapeutic approach.

A phase I multiple ascending-dose study in healthy subjects demonstrated favorable tolerability, with adverse events reported in approximately 35–45% of participants, most commonly mild upper respiratory tract infections, headache, and injection-site reactions [52]. No severe safety concerns were observed, and pharmacodynamic analyses confirmed sustained inhibition of CD40-dependent immune responses. Phase II clinical trials in SLE and lupus nephritis are ongoing, with preliminary results suggesting potential efficacy.

To date, no ocular adverse events have been reported in clinical trials or pharmacovigilance data. Nevertheless, experimental evidence implicates the CD40–CD40L axis in retinal vascular inflammation, microvascular dysfunction, and choroidal neovascularization [53,54,55]. Inhibition of this pathway could theoretically alter retinal immune homeostasis, potentially offering protection against inflammatory angiogenesis but at the cost of impaired immune surveillance within the eye. Such changes may predispose patients to infectious retinitis or subtle microvascular alterations not captured in early clinical studies.

Given these mechanistic considerations, the incorporation of structured ophthalmological endpoints into future trials is advisable. Advanced retinal imaging, including OCT, OCT angiography, and fundus photography, may help delineate the true ocular safety profile of CD40 blockade in SLE.

### 2.8. Janus Kinase (JAK) Inhibitors

Janus kinase (JAK) inhibitors are a novel class of oral immunomodulatory agents that interfere with intracellular signaling downstream of multiple cytokine receptors, including type I interferons, interleukins, and growth factors. Several JAK inhibitors have been investigated in systemic lupus erythematosus, including baricitinib (JAK1/2 inhibitor), tofacitinib (JAK1/3 inhibitor), and the selective TYK2 inhibitor deucravacitinib. Baricitinib demonstrated efficacy in reducing arthritis and cutaneous manifestations in the SLE-BRAVE trials, while deucravacitinib showed promising activity in early-phase studies [56,57].

The safety profile of JAK inhibitors in SLE appears consistent with that observed in rheumatoid arthritis and psoriasis. The most frequently reported adverse events include upper respiratory tract infections (10–20%), headache (8–12%), and gastrointestinal symptoms. Serious adverse events comprise opportunistic infections (notably herpes zoster, incidence 4–6%), venous thromboembolism (0.5–1.5%), and laboratory abnormalities such as cytopenias and hyperlipidemia [56,57]. Long-term extension studies in other autoimmune diseases further corroborate their overall manageable safety, but highlight the need for vigilance regarding infection risk [58,59,60].

With respect to ocular safety, no consistent drug-related ophthalmological toxicities have been documented in lupus trials. However, herpes zoster ophthalmicus has been described in patients receiving JAK inhibitors for other autoimmune conditions, reflecting impaired antiviral defense mechanisms [58]. Isolated case reports have also suggested associations with keratitis and scleritis, though causality remains uncertain. Given that JAK–STAT signaling is essential for interferon-mediated retinal immune surveillance, there is a biologically plausible risk of viral retinitis or reactivation of latent ocular infections during prolonged therapy.

In summary, JAK inhibitors are not intrinsically associated with direct ocular toxicity, in contrast to hydroxychloroquine or corticosteroids. Nonetheless, their immunomodulatory effects necessitate careful clinical vigilance, particularly for opportunistic viral eye infections. Incorporation of ophthalmological endpoints, including structured retinal imaging and viral retinitis surveillance, into future clinical trials would be valuable to establish the real-world ocular safety profile of this therapeutic class.

### 2.9. Cell-Based Therapies: Mesenchymal Stromal Cells (MSCs)

Mesenchymal stromal cells (MSCs) have emerged as a potential therapeutic option for refractory systemic lupus erythematosus and lupus nephritis, owing to their potent immunomodulatory properties, ability to restore regulatory T- and B-cell balance, and capacity to promote tissue repair. Both adipose-derived and umbilical cord-derived MSCs have been evaluated in early-phase clinical trials. A phase I study of allogeneic adipose-derived MSCs demonstrated encouraging clinical responses with acceptable tolerability in patients with refractory lupus nephritis [61]. Similarly, infusion of umbilical cord-derived MSCs in treatment-refractory SLE resulted in improved disease activity indices, reduction of autoantibody titers, and no dose-limiting toxicity [62].

The overall safety profile of MSC therapy has been favorable, with most adverse events consisting of mild infusion-related reactions such as fever, chills, headache, or transient hypertension, occurring in 10–20% of patients. Serious adverse events are rare (<5%) and are predominantly infection-related, reflecting the immunomodulatory rather than broadly immunosuppressive nature of MSCs [62]. Long-term data remain limited, but available studies suggest sustained efficacy without evidence of tumorigenicity or cumulative organ toxicity [61,63].

No ophthalmological complications have been directly reported in clinical trials of MSC therapy for SLE. Nevertheless, several theoretical risks warrant consideration. Systemic MSC infusion could potentially induce retinal microvascular occlusion or abnormal neovascularization due to endothelial interactions. In addition, modification of ocular immune privilege may predispose to immune dysregulation within the retina. Moreover, increased susceptibility to viral infections under MSC therapy raises the concern of opportunistic retinitis (e.g., cytomegalovirus, herpes simplex), particularly in patients with prior heavy immunosuppression.

At present, evidence does not support direct ocular toxicity from MSC therapy in SLE. However, given the eye’s vulnerability to both vascular injury and infection, incorporation of structured ophthalmologic endpoints—including fundus imaging, OCT, and viral retinitis surveillance—into future MSC trials would provide critical safety insights.

### 2.10. CD38 Inhibition: Mezagitamab (TAK-079)

Mezagitamab (TAK-079) is a fully human IgG1 monoclonal antibody targeting CD38, a glycoprotein highly expressed on plasma cells and activated immune subsets. By depleting autoreactive plasma cells, CD38 inhibition reduces pathogenic autoantibody production, making it an attractive therapeutic strategy in systemic lupus erythematosus. In a phase I study, mezagitamab demonstrated favorable safety, dose-dependent pharmacodynamics, and preliminary clinical efficacy, including reductions in anti-dsDNA titers and improvements in disease activity scores [64].

The overall safety profile was acceptable, with adverse events mainly consisting of infusion-related reactions (15–20%), mild infections (10–15%), and transient cytopenias (8–10%). No dose-limiting toxicities were observed, and importantly, no ophthalmological adverse events were reported in clinical trial datasets. However, ocular monitoring was not a predefined safety endpoint [64].

From a mechanistic perspective, potential ocular risks remain biologically plausible. CD38 is involved in NAD^+^ metabolism and regulation of oxidative stress in retinal tissues. Experimental data implicate CD38 in retinal neuroinflammation and photoreceptor homeostasis, suggesting that its inhibition may alter retinal resilience and increase susceptibility to neurodegenerative changes [65,66]. Moreover, systemic depletion of plasma cells can impair antiviral defenses, raising concern for opportunistic ocular infections such as cytomegalovirus (CMV) or herpes simplex virus (HSV) retinitis, particularly in heavily immunosuppressed patients.

At present, there is no evidence of direct ocular toxicity from CD38 inhibition in SLE. Nonetheless, considering the mechanistic associations with retinal metabolism and immune surveillance, incorporation of structured ophthalmological endpoints—including OCT, fundus imaging, and viral retinitis screening—into future clinical trials would be prudent to fully characterize the ocular safety profile of this therapeutic class.

### 2.11. Costimulatory Blockade: Iscalimab (Anti-CD40L)

Iscalimab is a fully human monoclonal antibody that selectively inhibits CD40 ligand (CD154), thereby blocking the CD40–CD40L costimulatory axis essential for B-cell activation, germinal center formation, and autoantibody production. In systemic lupus erythematosus, aberrant CD40–CD40L signaling plays a critical role in sustaining autoimmunity and mediating tissue damage, making it an attractive therapeutic target. A phase II randomized controlled trial in patients with proliferative lupus nephritis demonstrated acceptable safety, predictable pharmacokinetics, and preliminary efficacy signals [67].

The overall safety profile of iscalimab was favorable. The most common adverse events included upper respiratory tract infections (10–15%), headache (8–10%), and infusion-related reactions (6–8%). Serious adverse events were infrequent (<10%), and no new long-term safety concerns were identified. Importantly, no ocular adverse events have been reported in the available clinical studies. However, structured ophthalmological assessments were not incorporated into trial protocols, limiting conclusions on drug-specific ocular safety [67].

Mechanistic considerations suggest potential ocular implications. The CD40–CD40L pathway has been implicated in retinal vascular inflammation, pathological angiogenesis, and maintenance of endothelial integrity in experimental models [53,55]. Inhibition of this pathway may theoretically reduce angiogenic activity and protect against certain inflammatory ocular processes. Conversely, impaired immune surveillance could predispose patients to infectious retinitis or subtle microvascular dysfunction within the retina. These concerns remain speculative but underscore the need for systematic ophthalmologic evaluation.

In summary, while current evidence indicates a favorable safety profile for iscalimab with no direct ocular toxicity identified, the absence of standardized ophthalmologic endpoints remains a significant knowledge gap. Future trials should integrate multimodal retinal imaging (e.g., OCT, fluorescein angiography) to capture early microvascular or neuro-ophthalmic changes and establish a comprehensive ocular safety profile.

A consolidated summary of the therapeutic agents reviewed herein, including their clinical efficacy and ocular safety profiles, is presented in Table 1.

Overall, this analysis underscores the substantial heterogeneity in ocular safety profiles across therapeutic classes for systemic lupus erythematosus. Conventional agents, particularly hydroxychloroquine and systemic corticosteroids, exhibit the most consistent and well-characterized ocular risks, including irreversible retinopathy, posterior subcapsular cataract formation, steroid-induced glaucoma, and central serous chorioretinopathy. These complications are dose- and duration-dependent, frequently progressive, and remain major limitations in long-term management.

In contrast, newer biologics and targeted therapies—such as belimumab, anifrolumab, voclosporin, and dual BAFF/APRIL inhibitors—have not demonstrated definitive ophthalmic toxicity in randomized clinical trials. However, their mechanisms of action raise biologically plausible concerns, and isolated case reports highlight the need for ongoing pharmacovigilance. Importantly, the lack of structured ophthalmological endpoints in most pivotal trials represents a barrier to definitive safety conclusions.

Experimental and emerging therapies—including JAK inhibitors, anti-CD40/anti-CD40L antibodies, CD38-directed therapy, and mesenchymal stromal cell-based approaches—currently lack robust ophthalmological datasets. Their potential for subtle microvascular, inflammatory, or infection-mediated ocular complications remains theoretical but biologically plausible.

Taken together, these findings emphasize the importance of systematic ophthalmic monitoring across all therapeutic classes in SLE, both to mitigate well-established risks of conventional drugs and to proactively identify emerging safety signals with novel immunomodulatory agents. Integrating standardized retinal imaging modalities and functional visual assessments into future clinical trials would substantially strengthen the evidence base and help safeguard visual outcomes in this vulnerable patient population.

## 3. Discussion

The present review highlights the complexity of evaluating ocular safety in systemic lupus erythematosus therapeutics. While the introduction of biologic and targeted agents has transformed disease management, their ophthalmic safety profiles remain incompletely characterized. In contrast, conventional drugs—particularly hydroxychloroquine (HCQ) and systemic corticosteroids—are consistently linked to well-defined and clinically significant ocular toxicities that continue to represent a major source of morbidity.

### 3.1. Hydroxychloroquine and Corticosteroids as Benchmarks for Ocular Toxicity

HCQ is associated with a well-documented, dose- and duration-dependent risk of retinal toxicity, most classically presenting as bull’s-eye maculopathy with progressive outer retinal atrophy. Importantly, retinal damage may progress despite discontinuation of therapy, underscoring the importance of preventive screening and early detection [36,68]. Similarly, systemic corticosteroids, while indispensable for acute disease control, are strongly associated with posterior subcapsular cataracts, steroid-induced glaucoma, and central serous chorioretinopathy, which can result in irreversible vision loss if not identified promptly [69]. These agents therefore remain the reference standard against which ocular safety of emerging therapies must be assessed.

### 3.2. Novel Therapies and Emerging Safety Signals

Biologics such as belimumab and anifrolumab, along with voclosporin and dual BAFF/APRIL inhibitors, have not demonstrated consistent ocular toxicities in randomized clinical trials. Nevertheless, isolated reports describe inflammatory complications—including episcleritis, anterior uveitis, and neuroretinitis—as well as subtle microvascular alterations detected by optical coherence tomography angiography (OCTA) [2,7]. These changes may result from immune dysregulation within the retina or vascular endothelial dysfunction. Increased susceptibility to opportunistic infections during systemic immunomodulation could also play a role. In contrast, investigational approaches such as JAK inhibitors, anti-CD40/anti-CD40L therapies, CD38 blockade, and mesenchymal stromal cells (MSCs) remain insufficiently evaluated with respect to ocular outcomes, though biologically plausible risks include viral retinitis, retinal microangiopathy, and neuroretinal dysfunction.

Recent advances in retinal immunobiology have provided new insight into the potential mechanisms underlying ocular adverse events related to immune-targeted therapies [2,7]. The retinal immune microenvironment is maintained through tightly regulated interactions between Müller glia, microglia, and retinal endothelial cells, which collectively sustain immune privilege and neurovascular integrity. Disruption of these pathways—particularly the JAK–STAT and interferon signaling axes targeted by JAK inhibitors—may alter microglial activation and glial–vascular communication, predisposing to subclinical neuroinflammation or microvascular dysfunction [58]. Although no consistent clinical signal has yet been identified, these mechanistic findings justify prospective imaging studies using OCTA and retinal immunometabolic biomarkers to monitor for early changes during JAK inhibitor therapy.

Similarly, mesenchymal stromal cell (MSC) therapies exert complex immunoregulatory effects on the retinal environment through secretion of cytokines, exosomes, and growth factors. While these mechanisms are generally protective, excessive local immunosuppression or altered vascular signaling may theoretically disrupt retinal immune privilege. Experimental evidence indicates that MSC-derived exosomes modulate microglial activation and complement pathways, influencing local inflammatory balance and neurovascular stability [2,4,58]. Although clinical ophthalmic toxicity has not been observed, such immunometabolic modulation supports the need for targeted preclinical studies assessing retinal microenvironmental responses during cell-based immunotherapy.

### 3.3. Spectrum of Potential Ocular Complications

Across therapeutic classes, ocular adverse events in SLE can be broadly categorized into four domains:Inflammatory: episcleritis, anterior uveitis, scleritis, chorioretinitis, neuroretinitis;Vascular: retinal vasculitis, microaneurysms, capillary non-perfusion (OCTA-detected);Neuro-ophthalmic: optic neuritis, ischemic optic neuropathy, cranial nerve palsies;Structural: posterior subcapsular cataracts, steroid-induced glaucoma, CSCR, HCQ-related maculopathy.

Although many of these manifestations are rare, their potential severity and irreversibility justify systematic surveillance. Of note, subclinical microvascular changes may precede overt symptoms and could serve as early biomarkers of drug-related ocular toxicity [4,6].

### 3.4. Role of Ophthalmologic Monitoring in SLE Care

Given the spectrum of potential ocular complications, ophthalmological evaluation should be considered an integral component of SLE management. Baseline assessment is recommended for all patients prior to initiation of systemic therapy, with follow-up intervals tailored to drug exposure, duration of treatment, and patient-specific risk factors.

Core ophthalmological examinations include:Best-corrected visual acuity (BCVA): measured with standardized visual acuity charts (ETDRS or Snellen) under optimal refraction. BCVA provides a reproducible baseline of central visual function and enables longitudinal detection of subtle changes.Intraocular pressure (IOP) measurement: Goldmann applanation tonometry remains the gold standard, while rebound tonometry or non-contact tonometry may be used in selected settings. Regular IOP monitoring is essential to detect corticosteroid-induced ocular hypertension and secondary glaucoma.Dilated fundus examination: performed via pharmacological mydriasis and indirect ophthalmoscopy, or slit-lamp biomicroscopy with high-resolution fundus lenses (e.g., Volk 90D, 78D, or Goldmann three-mirror). This allows assessment of the optic disc, macula, and retinal vasculature, facilitating recognition of inflammatory, vascular, or neuro-ophthalmic complications.Spectral-domain optical coherence tomography (SD-OCT): a high-resolution, non-invasive modality that visualizes retinal microarchitecture. SD-OCT is the gold standard for early detection of hydroxychloroquine retinopathy and can detect parafoveal thinning before clinical symptoms appear.Fundus autofluorescence (FAF): captures natural lipofuscin-related fluorescence in the retinal pigment epithelium (RPE), allowing identification of early RPE dysfunction and toxic retinopathy at a preclinical stage.Optical coherence tomography angiography (OCTA): enables non-invasive, depth-resolved visualization of retinal and choroidal microvasculature. OCTA is particularly valuable for detecting subclinical ischemic changes, capillary dropout, and inflammatory microangiopathy in SLE patients [4,6].Visual field testing (10-2 protocol): automated perimetry targeting the central 10 degrees of vision. This functional test is highly sensitive for early paracentral scotomas associated with hydroxychloroquine toxicity and complements OCT.Fluorescein angiography (FA): an invasive but highly informative technique, considered the reference standard for diagnosing retinal vasculitis, vascular leakage, and neovascularization.

Multimodal imaging is particularly valuable because it enables detection of subclinical pathology. Early recognition of such changes can significantly influence therapeutic decisions. For example, structural alterations on OCT in patients receiving HCQ can justify early dose reduction, potentially preventing irreversible maculopathy. Likewise, OCTA may serve as a biomarker of microvascular changes in patients treated with biologics, offering insights into drug-related vascular effects before clinical symptoms arise.

In summary, systematic ophthalmologic monitoring—anchored in multimodal assessment—represents a cornerstone of safe long-term SLE management. Embedding these modalities into both clinical practice and therapeutic trials will not only reduce vision-threatening complications but also refine understanding of how systemic therapies influence ocular tissues. From a clinical standpoint, translating these principles into everyday practice can significantly enhance patient outcomes. Multimodal imaging and functional testing provide objective, reproducible tools for early detection of drug-induced or disease-related ocular changes, allowing timely treatment modification and prevention of irreversible damage. Incorporating standardized ophthalmologic assessments into follow-up schedules enables clinicians to personalize therapy intensity, anticipate adverse events, and safeguard visual function as an essential part of comprehensive lupus care.

### 3.5. Need for Interdisciplinary Collaboration

The findings of this review strongly support interdisciplinary collaboration as a cornerstone of optimal SLE care. Ocular complications are often underrecognized in routine rheumatology practice, while ophthalmologists may be less familiar with the evolving landscape of immunomodulatory therapies. To bridge this gap, structured collaboration is required at several levels:In clinical practice, joint rheumatology–ophthalmology clinics and standardized referral pathways help identify vision-threatening complications in time. Shared care protocols also facilitate risk-stratified monitoring.Clinical trials: incorporation of ophthalmological endpoints—including multimodal imaging, functional testing, and standardized adverse event reporting—into trial protocols will allow more accurate characterization of ocular safety profiles.Education and patient engagement: interdisciplinary teams can provide comprehensive education, empowering patients to recognize early visual symptoms and adhere to surveillance schedules. This is particularly relevant for HCQ-treated individuals, where delayed detection of retinopathy remains common.Research networks: collaborations between ophthalmology, rheumatology, nephrology, and immunology researchers will be essential to uncover mechanistic links between systemic immune dysregulation and ocular manifestations, and to evaluate potential drug–tissue interactions at the ocular level.

Such integrative approaches foster a patient-centered model of care, ensuring that systemic disease control is not achieved at the expense of long-term visual function.

In clinical practice, effective collaboration between rheumatology and ophthalmology should follow a structured pathway. Baseline ophthalmic evaluation is recommended before initiating immunomodulatory therapy, with follow-up visits every 6–12 months depending on cumulative drug exposure, ocular risk factors, and disease activity. Referral to ophthalmology should be mandatory in cases of new visual symptoms, persistent ocular surface inflammation, or systemic therapy escalation. Joint ward rounds or multidisciplinary clinics, where ophthalmologists and rheumatologists review complex cases together, can facilitate early recognition of toxicity signals and coordinated therapeutic adjustments. Such standardized referral criteria and communication channels enhance both patient safety and the continuity of care.

### 3.6. Future Perspectives

The current understanding of ocular safety profiles among emerging SLE therapies remains heterogeneous in terms of evidence quality. While conventional agents such as hydroxychloroquine and corticosteroids are supported by extensive Level I–II data, most novel immunomodulators and cell-based therapies rely on Level III–IV evidence derived from early-phase trials, observational studies, or mechanistic investigations. This disparity underscores the importance of long-term, standardized ophthalmic safety monitoring to accurately determine the true incidence and mechanisms of drug-related ocular events in real-world practice.

Ophthalmic safety evaluation should be structured according to the stage of drug development to ensure early identification and continuous monitoring of potential ocular effects. In preclinical and Phase I research, retinal toxicity models and exploratory imaging biomarkers such as OCT and OCTA can reveal subclinical changes before human exposure [2]. Phase II–III trials should integrate standardized ophthalmological endpoints and scheduled retinal assessments within composite safety outcomes, ensuring consistency across sites and study arms. In post-marketing and real-world settings, pharmacovigilance registries and digital ophthalmic databases remain essential to detect delayed or rare ocular events and validate biomarker correlations observed in earlier phases [18]. Such a tiered, phase-specific approach provides a translational framework for comprehensive ocular safety assessment throughout the therapeutic life cycle.

Although conventional therapies remain the predominant cause of ocular complications in SLE, novel immunomodulators may ultimately provide safer alternatives—if robust long-term surveillance confirms their ocular safety. Future directions should prioritize:Prospective longitudinal cohorts: large-scale, multicenter studies incorporating predefined ophthalmological endpoints are essential to capture subclinical changes and rare complications that are missed in short-term trials.Integration of imaging biomarkers with systemic markers: combining OCTA or FAF findings with serological or molecular biomarkers (e.g., interferon signatures, BAFF/APRIL levels) could help stratify risk and identify patients most vulnerable to drug-induced ocular toxicity.Artificial intelligence (AI) applications: recent studies have demonstrated that AI-based analysis of retinal images, including OCT and fundus photography, can accurately detect early signs of retinal pathology in autoimmune and drug-induced conditions. For example, deep learning models have been successfully applied to identify hydroxychloroquine retinopathy and subtle microvascular changes in systemic autoimmune diseases, supporting their potential as tools for automated toxicity screening and risk stratification. Integrating such validated approaches into longitudinal SLE monitoring frameworks could substantially enhance the sensitivity and efficiency of ophthalmological surveillance [70].Real-world data and registries: pharmacovigilance registries and electronic health record integration can provide critical post-marketing surveillance data, complementing randomized trial evidence and informing guideline development.Personalized surveillance strategies: precision medicine approaches may allow tailoring of ophthalmological monitoring intensity based on drug exposure, cumulative risk factors, and genetic predisposition.

Ultimately, the convergence of advanced imaging technologies, biomarker-driven risk stratification, and interdisciplinary collaboration offers the potential to redefine ocular safety assessment in SLE. By embedding ophthalmology into both clinical practice and therapeutic innovation pipelines, future research can safeguard visual outcomes while enabling full exploitation of novel immunomodulatory strategies.

## 4. Materials and Methods

This narrative review was conducted to synthesize available evidence on ocular adverse events associated with conventional and emerging therapies for systemic lupus erythematosus. The methodology was designed to ensure comprehensive coverage of the literature while maintaining a clinically relevant focus on ophthalmological safety.

### 4.1. Literature Search Strategy

A targeted literature search was performed in PubMed, complemented by manual screening of reference lists from relevant articles and recent reviews. The search covered publications from the past decade, focusing on peer-reviewed studies addressing SLE therapies and their ophthalmological safety. Search terms included combinations of systemic lupus erythematosus, ocular adverse events, hydroxychloroquine, corticosteroids, biologic agents, targeted therapies, and ophthalmic safety. Only peer-reviewed publications were considered to ensure reliability and scientific rigor.

### 4.2. Eligibility of Sources

Eligible studies included randomized controlled trials, observational cohort and case–control studies, case series, and case reports involving adult patients with a confirmed diagnosis of SLE. Interventions of interest encompassed both conventional immunotherapies (e.g., hydroxychloroquine, corticosteroids, azathioprine, mycophenolate, cyclophosphamide) and recently approved or investigational therapies (e.g., belimumab, anifrolumab, voclosporin, dual BAFF/APRIL inhibitors, JAK inhibitors, CD40/CD40L blockade, CD38 inhibition, and mesenchymal stromal cell-based approaches).

Studies were excluded if they were: (i) non-English publications, (ii) preclinical in vitro or animal studies, or (iii) clinical reports that lacked any ophthalmological outcomes.

### 4.3. Data Extraction and Organization

From each eligible study, the following data were extracted: therapeutic class, study design, patient population, intervention type, and reported ocular adverse events. To facilitate thematic analysis, findings were grouped by mechanism of action (e.g., B-cell targeted agents, interferon pathway inhibition, calcineurin inhibitors, conventional immunotherapies, and novel agents). Reported ocular complications were categorized into four broad domains: inflammatory, vascular, neuro-ophthalmic, and structural changes.

In addition to documented clinical adverse events, potential ophthalmological risks inferred from the pharmacological mechanism of action were also considered. This approach was adopted to highlight possible safety concerns that may not yet be evident in clinical practice or trial data but could plausibly arise from the targeted pathways of emerging therapies.

To enhance interpretative transparency, the strength of evidence was qualitatively graded according to study design hierarchy: randomized controlled trials (Level I), prospective or retrospective cohort studies (Level II), case series or pharmacovigilance data (Level III), and mechanistic or in vitro evidence (Level IV). The predominant evidence level for each therapeutic class is indicated in Table 1.

### 4.4. Data Presentation

Given the heterogeneity of study designs, treatment regimens, and outcome measures, the findings were synthesized using a qualitative narrative approach. Results are presented descriptively and organized by therapeutic category, with comparative references to conventional treatments for contextual understanding. Table 1 was developed to compile and categorize therapeutic classes according to their ocular safety profiles based on the reviewed literature.

## 5. Conclusions

The therapeutic armamentarium for SLE has expanded considerably with the introduction of targeted biologics and small-molecule immunomodulators. These advances hold promise for improved systemic disease control, but they also introduce new safety considerations, particularly in relation to ocular health—an area that remains underexplored. This review highlights that drug-induced ocular toxicity in SLE is mechanistically diverse, spanning inflammatory, vascular, neuro-ophthalmic, and structural pathways.

Among conventional agents, hydroxychloroquine and systemic corticosteroids remain the most consistent sources of ophthalmic morbidity, with well-established risks of irreversible retinopathy, posterior subcapsular cataracts, steroid-induced glaucoma, and central serous chorioretinopathy. In contrast, emerging biologics and investigational therapies—including belimumab, anifrolumab, voclosporin, dual BAFF/APRIL inhibitors, rituximab, JAK inhibitors, anti-CD40/anti-CD40L antibodies, CD38 blockade, and mesenchymal stromal cells—have not yet demonstrated comparable, reproducible ocular toxicities. Nonetheless, isolated reports and biologically plausible mechanisms suggest that subtle or infection-mediated complications may be underdetected.

These observations underscore the critical need for integrated ophthalmologic surveillance in SLE care. Baseline evaluation followed by longitudinal monitoring with sensitive modalities—such as SD-OCT, fundus autofluorescence, visual field testing, and OCTA—can enable detection of subclinical pathology before irreversible damage occurs. Surveillance strategies should be individualized, taking into account both the drug’s mechanism of action and patient-specific risk factors.

Looking ahead, ophthalmological endpoints must be embedded into clinical trials of novel SLE therapies. Prospective studies, mechanistic investigations, and real-world pharmacovigilance registries are essential to establish the true incidence, timing, and reversibility of ocular events. The integration of imaging biomarkers, patient-reported outcomes, and advanced analytic methods (e.g., artificial intelligence applied to retinal imaging) may further refine risk stratification and early detection strategies.

The practical implications of these findings extend beyond safety monitoring. Incorporating systematic ophthalmologic assessment into SLE management can improve real-world care by enabling early detection of at-risk patients and preventing irreversible visual complications. Establishing unified monitoring protocols and fostering close collaboration between ophthalmologists and rheumatologists may further enhance treatment safety and optimize long-term outcomes. Ultimately, these strategies align with the principles of precision medicine—translating therapeutic advances into individualized, safer, and patient-centered lupus care.

In conclusion, preserving vision should be regarded as a core component of comprehensive SLE management. Strengthening interdisciplinary collaboration between rheumatologists, ophthalmologists, and clinical researchers will be essential to ensure that the benefits of innovative therapies are realized without compromising ocular health. Safeguarding visual function is not only vital for patient quality of life but also central to achieving holistic therapeutic success in systemic lupus erythematosus.

## Figures and Tables

**Table 1 ijms-26-11744-t001:** Summary of systemic lupus erythematosus (SLE) therapies and associated ocular adverse events.

Therapy	Efficacy in SLE	Most Frequent AEs	Serious AEs	Ocular AEs (Documented/Potential)	Predominant Evidence Level
Belimumab (anti-BAFF) [8,9,10,11,12,13]	↓ disease activity; effective in lupus nephritis	URTI, headache, nausea	Infections (~5%), rare malignancies	None consistent; vigilance advised	I
Anifrolumab (anti-IFNAR1) [15,16,17,18,19,20,21]	Improves global activity; allows steroid tapering	URTI, bronchitis, infusion reactions, herpes zoster	Herpes zoster (5–7%), pneumonia	None documented; theoretical retinal immune effects	I
Voclosporin (calcineurin inhibitor) [22,23,24,25,26,27]	↑ renal response in lupus nephritis	GI upset, hypertension, headache, renal effects	Infections (~21%), renal toxicity, hypertension	None confirmed; class effect: ↑ IOP, cataracts (possible)	I
Telitacicept/Atacicept (dual BAFF/APRIL inhibitors) [26,27,28,29]	Telitacicept: ↑ SRI-4 response; Atacicept halted (safety)	URTI, nasopharyngitis, headache, injection-site reactions	Infections (8–12%), hypogammaglobulinemia (atacicept)	None confirmed; blurred vision (isolated cases)	II
Hydroxychloroquine (HCQ) [35,36,37,38,39,40,41,42]	Maintains remission, ↓ flares, ↑ survival	Retinal toxicity	Retinopathy prevalence: 7.5% (>5 yrs), >20% (>20 yrs)	Definite: bull’s-eye maculopathy, retinal atrophy	I–II
Corticosteroids [43,44,45,46]	Rapid control, life-saving in severe flares	Weight gain, hypertension, hyperglycemia	Cataracts (20–40%), glaucoma (18–36%), CSCR (OR >4)	Definite: PSC cataracts, steroid glaucoma, CSCR	I–II
Rituximab (anti-CD20) [47,48,49,50,51]	Benefit in refractory SLE/lupus nephritis (registries)	Infusion reactions, infections	Serious infections (10–15%), viral reactivation	Rare: CMV/HSV retinitis (opportunistic)	II–III
Anti-CD40 (BI 655064) [52,53,54,55]	Early-phase; potential efficacy	URTI, headache, injection-site reactions	No major SAE in phase I	None reported; possible retinal vascular effects	II–IV
JAK inhibitors (baricitinib, tofacitinib, deucravacitinib) [56,57,58,59,60]	Baricitinib effective for joints/skin; TYK2 promising	URTI, headache, GI upset, herpes zoster	Opportunistic infections, VTE, herpes zoster ophthalmicus	HZ ophthalmicus, keratitis; viral retinitis possible	I–II
Mesenchymal stromal cells (MSCs) [61,62,63]	Phase I/II: benefit in refractory SLE/lupus nephritis	Infusion reactions, mild infections	Rare serious infections (<5%)	None reported; theoretical microvascular occlusion, retinitis	II–III
CD38 inhibitors (mezagitamab) [64,65,66]	Phase I: ↓ autoantibodies, improved activity	Infusion reactions, mild infections, cytopenias	Cytopenias, infections	None confirmed; possible retinal NAD^+^/immune effects	II–IV
Anti-CD40L (iscalimab) [53,55,67]	Phase II in lupus nephritis: preliminary efficacy	URTI, headache, infusion reactions	Infections (<10%)	None reported; theoretical retinal microangiopathy	II–III

Evidence level: I—randomized controlled trials or meta-analyses; II—cohort or long-term observational studies; III—case series or registries; IV—mechanistic or preclinical evidence. Abbreviations: AE—adverse event; SAE—serious adverse event; URTI—upper respiratory tract infection; IOP—intraocular pressure; CSCR—central serous chorioretinopathy; BAFF—B-cell activating factor; APRIL—a proliferation-inducing ligand; SRI-4—SLE Responder Index 4; TYK2—tyrosine kinase 2; VTE—venous thromboembolism; HZ—herpes zoster; ↑ indicates increase; ↓ indicates decrease.

## Data Availability

No new data were created or analyzed in this study. Data sharing is not applicable to this article.

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
