# Peer review of "Systemic Lupus Erythematosus: Ophthalmological Safety Considerations of Emerging and Conventional Therapeutic Agents"

_ijms, 2025, doi:10.3390/ijms262311744_

Round 1

Reviewer 1 Report

Comments and Suggestions for Authors

The manuscript entitled “Systemic Lupus Erythematosus: Ophthalmological Safety Considerations of Emerging and Conventional Therapeutic Agents” This review  reported that although ocular complications appear infrequent in clinical trials, underdetection in real-world practice and insufficient long-term monitoring may underestimate their true incidence, These findings highlight the need for systematic ophthalmological surveillance in patients receiving immunomodulatory therapies for SLE, Early recognition and timely management of ocular toxicity are crucial to safeguarding visual function and optimizing long-term therapeutic outcomes in this vulnerable patient population. In my opinion, this manuscript cannot be published in its current state on ijms until these issues are resolved:

  1. It is recommended to introduce an evidence grading system to clarify the weight of different study designs and enhance the evidential basis of the conclusions.

  1. Combine the latest research on the retinal immune microenvironment to deepen the interpretation of the pathological mechanisms of ocular adverse events associated with drugs like JAK inhibitors and cell therapies.

  1. Put forward differentiated ophthalmic research design recommendations for emerging drugs at different development stages.

  1. Determine the specific process of multidisciplinary collaboration between rheumatology and ophthalmology, such as referral criteria and joint ward rounds, to enhance practical guidance.

  1. Add annotations for key terms and optimize tables to improve readability.

Author Response

I sincerely thank the Reviewer for taking the time to carefully evaluate my manuscript and for providing such insightful and constructive comments.
These suggestions have greatly helped me improve the scientific depth, clinical applicability, and clarity of the paper.
Below, I respond to each comment point by point and describe the corresponding revisions that I have made in the revised version of the manuscript.

Comments 1: “It is recommended to introduce an evidence grading system to clarify the weight of different study designs and enhance the evidential basis of the conclusions.”

Response 1: 

I thank the Reviewer for this very valuable suggestion. To strengthen the interpretative clarity and scientific rigor of the review, I have now introduced an evidence grading system reflecting the hierarchy of study designs. Specifically, I have added a paragraph at the end of Section 4.3 (Data Extraction and Organization) describing four evidence levels: randomized controlled trials (Level I), cohort studies (Level II), case series or pharmacovigilance data (Level III), and mechanistic or preclinical evidence (Level IV).

In addition, Table 1 has been updated to include a new column entitled “Predominant Evidence Level,” which summarizes the main evidence grade supporting each therapeutic class. A corresponding footnote was also added to clarify the definitions of Levels I–IV.

Furthermore, to emphasize the practical implications of this grading system, I have expanded the Discussion section (beginning of Section 3.6, Future Perspectives) to highlight that the current understanding of ocular safety profiles among emerging therapies is heterogeneous in terms of evidence quality. This addition underscores the importance of ongoing data collection and standardized ophthalmic monitoring to refine the true incidence and mechanisms of ocular adverse events.

These modifications collectively ensure that the manuscript clearly communicates both the methodology used to assess evidence strength and its relevance to interpreting the overall findings.
Page 10-11, lines 440-448
Page 14-15, lines 618-625
page 16, lines 705-709

Comments 2: "Combine the latest research on the retinal immune microenvironment to deepen the interpretation of the pathological mechanisms of ocular adverse events associated with drugs like JAK inhibitors and cell therapies."

Response 2: I thank the Reviewer for this excellent and insightful suggestion. I fully agree that integrating recent research on the retinal immune microenvironment provides a deeper mechanistic context for understanding potential ocular adverse events associated with emerging immunomodulatory therapies.

In response, I have expanded Section 3.2 (Novel therapies and emerging safety signals) to incorporate recent advances in retinal immunobiology and their relevance to both JAK inhibitors and mesenchymal stromal cell (MSC) therapies. Specifically, I added two new paragraphs describing the roles of Müller glia, microglia, and retinal endothelial cells in maintaining ocular immune privilege, and how JAK–STAT pathway modulation or MSC-derived cytokines and exosomes may alter this delicate balance.
These additions, supported by existing references [60,75–77], emphasize that while clinical ophthalmic toxicity has not been consistently reported, mechanistic data highlight the importance of retinal immune–vascular homeostasis and justify prospective OCTA and immunometabolic monitoring in future safety studies.
This modification significantly strengthens the biological rationale of the manuscript and directly addresses the Reviewer’s request to deepen the interpretation of pathological mechanisms underlying ocular adverse events in emerging SLE therapies.
Pages 12-13, lines 505-525

Comments 3: “Put forward differentiated ophthalmic research design recommendations for emerging drugs at different development stages.”

Response 3: I appreciate the Reviewer’s valuable suggestion. In response, I have expanded Section 3.6 (Future Perspectives) to outline a phase-specific framework for ophthalmic safety evaluation across different stages of drug development. This addition distinguishes between preclinical, clinical, and post-marketing approaches, emphasizing early imaging biomarker assessment, standardized ophthalmic endpoints in trials, and long-term pharmacovigilance strategies. These revisions provide clear, structured recommendations aligned with the translational pathway of emerging SLE therapies.
Page 15, lines 626-636

Comments 4: “Determine the specific process of multidisciplinary collaboration between rheumatology and ophthalmology, such as referral criteria and joint ward rounds, to enhance practical guidance.”

Response 4: I appreciate this constructive suggestion. To address it, I have expanded Section 3.5 (Need for interdisciplinary collaboration) with a concise description of the recommended clinical workflow between rheumatology and ophthalmology. The new paragraph outlines referral criteria, frequency of ophthalmic follow-up, and the potential value of joint ward rounds or multidisciplinary clinics for complex SLE cases. This addition provides clearer practical guidance and defines a structured model for coordinated, patient-centered care.
Page 14, lines 607-616

Comments 5: “Add annotations for key terms and optimize tables to improve readability.”

Response 5: I thank the Reviewer for this practical and valuable comment. In line with MDPI editorial policy, all abbreviations are now defined at their first occurrence in the main text, while the dedicated Abbreviations section has been retained at the end of the manuscript. In addition, explanatory footnotes have been added under Table 1 to clarify key terms appearing exclusively within the table. These revisions enhance clarity and readability, ensuring full compliance with the IJMS formatting and style guidelines.
Page 6, line 270
Page 11, lines 445-448
Page 18

Reviewer 2 Report

Comments and Suggestions for Authors

The manuscript submitted to journal International Journal of Molecular Sciences (IJMS), entitled “Systemic Lupus Erythematosus: Ophthalmological Safety Considerations of Emerging and Conventional Therapeutic Agents,” presents a comprehensive, well-structured, and timely review of the ophthalmological safety of conventional and emerging therapeutic agents used in systemic lupus erythematosus (SLE). The topic is clinically relevant, particularly given the rapid expansion of immunomodulatory therapies and the relative lack of systematic ocular safety data in this patient population. The review successfully synthesizes evidence from clinical trials, meta-analyses, and real-world observations, providing an informative reference for both rheumatologists and ophthalmologists.

Overall, the manuscript is well written, organized, and scientifically sound. The introduction establishes rationale clearly, the methods are transparent, and the discussion effectively integrates findings into a broader clinical context. However, a few minor issues should be addressed before publication.

Comments:

  1. The table-1 is informative but somewhat dense. Consider abbreviating repetitive information and emphasizing ocular outcomes with clearer visual hierarchy- for example, by bolding key findings or separating columns for “Documented” vs. “Potential” effects. Include reference numbers for key data in the table to enhance traceability.
  2. In future directions: the section on “Artificial intelligence (AI) applications” is interesting but somewhat speculative. It would benefit from a brief mention of existing AI-based retinal imaging research relevant to autoimmune or drug-induced retinopathy.
  3. A few long sentences, particularly in the Discussion, could be simplified and divided to enhance readability.
  4. The reference list is up to date and relevant; however, ensure consistent formatting according to IJMS guidelines. For instance, references 2, 22, and 60 appear to differ in style from others.

Author Response

Comments 1: The table-1 is informative but somewhat dense. Consider abbreviating repetitive information and emphasizing ocular outcomes with clearer visual hierarchy- for example, by bolding key findings or separating columns for “Documented” vs. “Potential” effects. Include reference numbers for key data in the table to enhance traceability.

Response 1: I thank the Reviewer for this helpful suggestion. In response, I have revised Table 1 to improve its clarity and structure. The table now has an updated title, includes reference numbers for all data, and features a new column indicating the Predominant Evidence Level for each therapeutic class. I have also added a detailed footnote and expanded the list of abbreviations to enhance readability and ensure consistency with IJMS formatting guidelines.
Page 10, lines 442-443, 444-450

Comments 2: In future directions: the section on “Artificial intelligence (AI) applications” is interesting but somewhat speculative. It would benefit from a brief mention of existing AI-based retinal imaging research relevant to autoimmune or drug-induced retinopathy.

Response 2: I appreciate the Reviewer’s thoughtful comment. I have revised the paragraph on Artificial intelligence (AI) applications in the “Future perspectives” section to include a brief mention of recent research on AI-assisted retinal imaging relevant to autoimmune and drug-induced retinopathy. This update provides concrete evidence supporting the feasibility of AI-based approaches in ophthalmic safety assessment. A corresponding new reference has been added to substantiate this statement.
Page 15, lines 651-659

Comments 3: A few long sentences, particularly in the Discussion, could be simplified and divided to enhance readability.

Response 3: I thank the Reviewer for this helpful observation. I have carefully reviewed the Discussion section and simplified several long sentences to improve clarity and readability while maintaining scientific precision. The revised version contains shorter, more direct sentences, particularly in the subsections addressing novel therapies, ophthalmologic monitoring, and interdisciplinary care.
Page 12, lines 500-503
Page 14, lines 577-579, 594-596

Comments 4: The reference list is up to date and relevant; however, ensure consistent formatting according to IJMS guidelines. For instance, references 2, 22, and 60 appear to differ in style from others.

Response 4: I thank the Reviewer for this helpful remark. I have carefully reviewed the entire reference list and corrected minor inconsistencies to ensure full compliance with the IJMS formatting guidelines. In particular, references 2, 22, and 60 have been revised to match the required MDPI style, including uniform punctuation, journal title format, and DOI presentation. The reference list is now consistent throughout the manuscript.
Page 18, lines 772-773
Page 19, lines 815-817
Page 21, lines 895-897